# Self-Assessment of Lower Urinary Tract Condition in Female Competitive Cyclists

**DOI:** 10.3390/healthcare12121163

**Published:** 2024-06-07

**Authors:** Mariola Saulicz, Aleksandra Saulicz, Edward Saulicz

**Affiliations:** 1Institute of Physiotherapy and Health Sciences, The Jerzy Kukuczka Academy of Physical Education, 40-065 Katowice, Poland; 2School of Public Health & Social Work, Queensland University of Technology (QUT), Kelvin Grove, QLD 4059, Australia

**Keywords:** women, bicycle riding, competitive sport, lower urinary tract symptoms

## Abstract

During cycling, prolonged compression by the bicycle saddle on the anatomical structures located in the perineum area occurs. An additional factor that may have a negative impact on organs located in the pelvic area may be a prolonged sitting position resulting in increased intraabdominal pressure. This situation has the potential to adversely affect pelvic floor function. Therefore, the aim of this study was to assess the incidence of lower urinary tract symptoms (LUTSs) in female competitive road cyclists and cross-country cyclists. The study included 76 female competitive road cyclists and cross-country cyclists and 76 women not practising competitive sport. The Core Lower Urinary Tract Symptom Score (CLSS) questionnaire was used to assess the lower urinary tract condition. Female competitive cyclists had a statistically significantly higher LUTSs score (95% CI: 3.12–4.2 vs. 2.31–3.16; *p* < 0.05) compared to women not practising competitive sports. Female cyclists had a statistically significantly higher overall CLSS score (95% CI: 3.99–5.61 vs. 2.79–3.97; *p* < 0.05). Female cyclists had a statistically significantly higher incidence and severity of urinary frequency (*p* < 0.05 and *p* < 0.02), urge (*p* < 0.001 and *p* < 0.02) and stress incontinence (*p* < 0.001 and *p* < 0.001), and pain in the bladder (*p* < 0.01 and *p* < 0.01), while physically inactive women recorded a statistically higher incidence of slow urinary stream (*p* < 0.01 and *p* < 0.04). A statistically significant association was recorded between the years of cycling and the number of hours per week spent on training and the number of symptoms and their severity. The number of natural births experienced by women involved in competitive cycling significantly affects the severity of LUT symptoms. Compared to women not practising competitive sports, competitive female cyclists are found to have a higher prevalence of LUTSs and a greater degree of severity. LUTSs in competitive female cyclists are negatively influenced by years of competitive career and weekly number of training hours and the number of natural births experienced.

## 1. Introduction

Cycling is not only a popular and economical form of transport, but it is also a simple form of recreational physical activity. Cycling as a means of transport and as an active leisure activity is also willingly practised by women. Studies show that in Denmark, Germany, and the Netherlands around half of all cycling trips are made by women, and in the United States of America, Canada and the United Kingdom this proportion is 25% [1,2]. Competitive forms of cycling are also becoming increasingly popular among women. Cycling, like any other form of sporting activity, is also fraught with potential adverse health effects. In addition to the typical injuries resulting from a fall or crash, injuries to the perineal area associated with the mechanical action of the bike saddle are typical for cycling [3]. Other risks are associated with prolonged pressure on the perineum by the bicycle saddle. This can result in symptoms of pain and tenderness of the buttocks, chafing, perineal folliculitis, furuncles, lymphedema and perineal numbness [2,3,4,5,6]. Perineal numbness and burning pain in the same area are caused by entrapment and compression of the pudendal nerve between the pubic bone and the saddle [2,7]. The pudendal nerve deriving its fibres from the S2–S4 segments is a mixed motor-sensory nerve innervating the genitalia and perineal muscles. The trunk of this nerve runs around the sacrospinous ligament, heads to the ischiorectal fossa and runs through the Alcock’s canal, where free gliding of this nerve is possible. A potential mechanism for some of the symptoms associated with the cycling position on the saddle and the pressure of the saddle itself on the perineum on the conduction in the pudendal nerve was previously presented [3]. Continuous pressure on the perineum by the bicycle saddle interferes with the gliding of this nerve within the Alcock canal, and the cyclist’s characteristic flexed trunk position associated with pedalling movements through the lower limbs results in the stretching of the pudendal nerve in the area of the tight sacrospinal and sacrotuberous ligaments, which promotes continuous overstretching of this nerve [3]. Such a condition would therefore result in a microneuropathy-like impairment of nerve conduction [8]. Another potential negative effect on the perineum is exerted by prolonged forced maintenance of the trunk flexion position. Low positioning of the handlebars in relation to the saddle results in increased perineal pressure and decreased genital sensation [9]. In addition, trunk flexion increases intra-abdominal pressure, which potentially negatively affects pelvic floor muscle tone and results in a higher risk of urinary incontinence (UI) symptoms [10].

Only a few studies in the literature consider symptoms of urological dysfunction among competitive female cyclists. LaSalle et al. [11] found that 19% of female cyclists had symptoms of haematuria/dysuria. Another study found a statistically significant history of urinary tract infections among female cyclists compared to non-cycling women [12]. In other studies, competitive female cyclists are only included in the studies on the prevalence of UI symptoms.

Research on the relationship between women’s competitive cycling and their urogenital health is rare. Studies focus mainly on the impact of maintaining a sitting position for a long time and the pressure of the saddle on the perineum. So far, researchers’ interest has focused predominantly on the occurrence of sexual disorders in competitive female cyclists. The occurrence of lower urinary tract symptoms (LUTSs), if included in these studies, was treated marginally and was limited to single symptoms. Due to the lack of data covering a wide range of LUTSs in women competitive cyclists, the aim of our study is to estimate the frequency of their occurrence. It was also decided to assess the relationship between the length of sports career and the intensity of weekly training and the severity of symptoms. Since some professional cyclists decide to become mothers during their sports careers, the relationship between the number and type of deliveries and the severity of LUTSs was analysed. To our knowledge, this is the first study to consider a broad spectrum of LUTSs among female competitive cyclists.

## 2. Materials and Methods

### 2.1. Study Design and Subjects

This was an observational, cross-sectional study conducted from autumn 2018 to autumn 2022. In 2020–2021, the study was conducted only during periods when there were no restrictions related to the COVID-19 pandemic. The study was conducted in Poland, in sports clubs that have a road cycling and cross-country cycling sections. Women for the control group were recruited from among women not involved in competitive sport and not undertaking regular recreational physical activity.

Ethical approval for the study was granted by the Bioethics Committee for Scientific Studies at the Jerzy Kukuczka Academy of Physical Education in Katowice (No. 6/2015). All study protocols were carried out in accordance with the Helsinki Declaration of Human Rights of 1975, modified in 1983. Each participant provided informed consent to partake in the study subsequent to receiving detailed information regarding its objectives and methodologies. The study was fully anonymous and at no stage of the study, i.e., collection of the results and their analysis, was it possible to identify the respondent who filled in the questionnaire.

Inclusion criteria among the female competitive cyclists were the age of at least 18 years and involvement in competition in road or cross-country cycling for at least 1 year. Pregnancy or a history of childbirth within the past year was an exclusion criterion in both the female athlete group and the control group. Among the female cyclists, no other health inclusion or exclusion criteria were applied due to the fact that the study was conducted on the premises of sports clubs on the occasion of training or sports competitions and during sports competitions. According to Polish law, fully healthy female athletes are allowed to participate in competitive sporting activities (training, competitions). However, only healthy women qualified for the control group, i.e., women who have not been diagnosed with any chronic disease, who have not had any infections in the past six months and who have not taken any pharmacological agents with a potential impact on lower urinary tract function in the past six months. Due to the inconclusive findings regarding the effect of running on UI, women who occasionally run recreationally in their leisure time were also excluded from the study [13,14]. Women who did not speak the Polish language to the extent that they could understand and consciously complete the questionnaire were also excluded.

In the first stage of the research, a short interview was conducted among professional cyclists, in which they were first asked for their consent to take part in the research. After obtaining acceptance, questions were asked about age, period of practising competitive cycling and any previous births. A total of 95 sets of questionnaires were distributed among professional cyclists who met the criteria for participation in the research (age at least 18 years old, practising competitive cycling for at least one year and no births in the last year). Of this number, 8 sets were not returned, and in 11 cases the Core Lower Urinary Tract Symptom Score (CLSS) questionnaires [15] were incomplete or basic demographic data (e.g., age) was missing or data regarding sports career was not provided. All in all, fully completed questionnaires were collected from 76 female competitive road cyclists and cross-country cyclists. The control group was recruited among young women who met similar criteria in terms of age and previous births, as among the professional cyclists. In addition, questions were asked about health status (occurrence of chronic diseases, infection within the last six months, and taking medication) and the level and type of physical activity. After obtaining negative answers regarding health and excluding recreational running, sets of questionnaires were distributed. Recruitment of the control group was completed after obtaining 76 completely completed questionnaires. The characteristics of the study groups and comparisons of the homogeneity of the groups are given in Table 1. In the group of competitive cyclists, 41 women were involved in road cycling and 35 women were involved in cross-country cycling.

The research was conducted using anonymous questionnaires. Women cyclists were selected from all over Poland. The questionnaires were distributed during training, sports competitions and training camps. Printed, anonymous questionnaires in the form of a stapled set consisting of a cover letter explaining the circumstances and purpose of the study, a questionnaire with socio-demographic data and several survey forms (Baecke, SF-36, FKB-20 and KCS [16,17,18,19]), among them a survey form assessing the occurrence of LUTSs, were distributed in a white stamped envelope with a return address or in a white unstamped envelope when the questionnaires could be returned to a collection container for the completed questionnaires. The LUTS assessment questionnaire was consciously placed in the questionnaire set, and the cover letter indicated self-assessment of health-oriented quality of life for female athletes as the aim of the study, in order to not overly focus the attention of the women surveyed on lower urinary tract issues. For this reason, the questionnaire assessing LUTSs was never placed first in the set of questionnaires, and after every 5th set, it was reordered by one place in the set.

The research did not specify the time required to complete the surveys—the surveys could be returned on the same day, the next day or returned by post.

### 2.2. Data Collection

The CLSS questionnaire [15] was used to assess the presence of LUTSs. The CLSS questionnaire was designed to be self-administered by the examined subject and assessed the presence of the most significant symptoms related to bladder and lower urinary tract function. The CLSS questionnaire took the form of a one-page printed table, which consisted of a total of 11 questions. The questionnaire contained 10 questions on lower urinary tract symptoms, i.e., increased frequency of urination during the day (question 1), nocturia—frequency of urination at night (question 2), sudden and strong urge to urinate (question 3), urge to urinate due to inability to hold back (question 4), urinating while coughing, sneezing or straining (Q5), urinating in a slow stream (Q6), urinating with effort (Q7), feeling incomplete emptying of the bladder after urination (Q8), bladder pain (Q9) and urethral pain (Q10). With the exception of the first 2 questions containing the number ranges of daily (question 1) and nocturnal (question 2) micturitions, in the remaining 8 questions, each answer related to the frequency of a given lower urinary tract symptom was given the score from 0 to 3, where 0 meant a negative answer of “no”, 1 meant “rarely”, 2 “sometimes” and 3 “often”. In question 1, 0 points were awarded for ~7 micturitions, 1 point for 8–9 micturitions, 2 points for 10–14 micturitions and 3 points for ≥15 micturitions. In question 2, 0 points were awarded for no nocturnal micturitions, 1 point for 1 nocturnal micturition, 2 points for 2–3 micturitions and 3 points for ≥4 nocturnal micturitions. Completion of all responses allowed the total score to be calculated for the frequency of each symptom, and the score obtained was 0 pts. (no symptoms of LUTS at all) to 30 pts. (maximum intensity of LUTS). In the final 11th question, the respondents were asked to rate their sense of satisfaction with their current lower urinary tract-related condition and the question read as follows: “If you were to spend the rest of your life with your urinary condition just the way it is now, how would you feel about that?”. There were 7 response options to choose from: Delighted (0 points), Pleased (1 point), Mostly satisfied (2 points), Mixed, equally satisfied and dissatisfied (3 points), Mostly dissatisfied (4 points), Unhappy (5 points), Terrible (6 points) [15].

### 2.3. Statistical Analysis

The characteristics of the participants were described by the mean and standard deviation and minimum and maximum values. The occurrence of lower urinary tract symptoms was presented numerically and by percentage within groups. The occurrence of symptoms and their severity were described by the mean, standard deviation and 95% CI. Differences in demographic parameters (age, height, weight, BMI) and scores of CLSS were analysed by *t*-test for independent samples or by the Mann–Whitney test, since data did not show normal distribution. Differences in symptoms and number of childbirths were analysed by the Chi^2^ test. Relationships between the years of sports career, training frequency and training hours per week, number and type of births experienced and CLSS scores were calculated using the Pearson’s linear correlation test. However, in relation to the answers to the detailed questions of the CLSS questionnaire, due to their rank nature (0–1–2–3), these relationships were calculated using the non-parametric Spearman’s rank correlation test. The level of significance was set at *p* < 0.05. Data were analysed using STATISTICA 13.3 PL (StatSoft Inc., Tulsa, OK, USA).

## 3. Results

The largest number of female competitive cyclists indicated the presence of one (13 persons; 17.1%) or three symptoms (13 persons; 17.1%) from the lower urinary tract (Figure 1). The maximum number of symptoms—10—was shown by one person (1.3%). Only four persons (5.3%) showed no symptoms of LUTS. In the control group, the highest percentage of respondents (25%) indicated the presence of three symptoms. Six female respondents in this group (7.9%) did not indicate the presence of any symptom from the lower urinary tract. Female competitive cyclists were statistically significantly more likely to indicate the presence of LUTSs (95% CI: 3.12–4.2 vs. 2.31–3.16) (Table 2). In contrast, there were no statistically significant differences between the female cyclist group and the control group in the level of acceptance of the condition of CLSS (95% CI: 0.95–1.57 vs. 0.8–1.28).

Of the 10 symptoms analysed, 7 occurred more frequently in competitive female cyclists and 3 in the control group (Figure 2). Four symptoms (daytime urinary frequency, urgency incontinence, stress incontinence and bladder pain) were statistically significantly more frequent in female cyclists. In contrast, slow-stream urination was more common in women in the control group.

Table 3 shows the mean values and their standard deviations as well as the 95% confidence interval of individual lower urinary-tract symptoms assessed in the CLSS questionnaire among professional cyclists and women from the control group. The greatest severity of the symptoms was related to stress incontinence and bladder pain and was found in women who were competitive cyclists (Table 3). Symptoms of stress incontinence occurred rarely in 16 female cyclists (21.05%), sometimes in 20 female cyclists (26.32%) and frequently in 2 (2.63%). On the other hand, bladder pain was infrequent in 24 female cyclists (31.58%), occasional in 12 female cyclists (15.79%) and frequent in 4 (5.26%). The symptom intensity levels of daytime urination frequency, urgency incontinence, stress incontinence and bladder pain were statistically significantly higher among the competitive female cyclists. In contrast, slow-stream urinary problems were statistically significantly more severe in the control group. The overall CLSS score was statistically significantly higher in the group of female competitive cyclists (95% CI: 3.99–5.61 vs. 2.79–3.97).

The majority of women in both study groups were positive about their current lower urinary tract health (73.68% in the competitive cycling group and 90.78% in the control group) (Figure 3). However, the percentage of respondents whose feelings about the condition of their lower urinary tract were at least mixed was higher in the female competitive cyclists (26.32% vs. 9.22%). Differences in feelings of satisfaction with health related to the current condition of the lower urinary tract were found to be statistically significant.

A statistically significant relationship between all correlated variables related to sports career, i.e., years of cycling, number of training sessions per week and number of training hours per week was found only in the case of the severity of symptoms of stress urinary incontinence (Table 4). Longer years of competitive cycling, a greater number of training sessions per week and a greater number of hours per week devoted to cycling training are statistically significantly associated with greater severity of symptoms of stress urinary incontinence. In relation to the years of sports career and the number of training sessions per week, there was a weak correlation (R = 0.379; *p* < 0.001 and R = 0.247; *p* < 0.05, respectively), while in relation to the number of training hours, a moderate correlation was found (R = 0.437; *p* < 0.001). The overall condition of the lower urinary tract, expressed in the total value of the questionnaire, is influenced to some extent by years of cycling and the number of hours per week devoted to cycling training. In both cases, a weak relationship was recorded (r = 0.364; *p* < 0.001 and r = 0.288; *p* < 0.01, respectively). In relation to the remaining analysed variables, single statistically significant relationships were noted, and the recorded correlation coefficients indicated a weak correlation.

Of the 10 lower urinary tract symptoms analysed, the number of natural deliveries among competitive female cyclists was not statistically significantly associated with only three symptoms (urgency, pain in the bladder and pain in the urethra) (Table 5). The strongest association was registered with symptoms of stress incontinence (R = 0.517; *p* < 0.001) and it was a moderate correlation. A moderate correlation between the number of natural deliveries was also observed in relation to nocturia (R = 0.427; *p* < 0.001) and slow urinary-stream symptoms (R = 436; *p* < 0.001). In the case of the remaining four symptoms of LUTSs that statistically significantly correlated with the number of natural births, these were weak correlations. In the control group, a statistically significant association was recorded only between past natural deliveries and symptoms of stress incontinence and this correlation was weak (r = 0.288; *p* < 0.05). In the group of competitive female cyclists, past natural deliveries directly correlated with the number of symptoms (R = 0.542; *p* < 0.001) and the overall symptom severity index (r = 0.569; *p* < 0.001). Therefore, the general condition of LUTS shows a moderate relationship with previous births in women practising cycling competitively. No such associations were registered among women in the control group. No association was found between the number of pregnancy terminations by caesarean section in the two study groups.

## 4. Discussion

In the source literature, only one paper on the prevalence of LUTSs among female athletes included female cyclists. In other reports on lower urinary tract symptoms where female competitive cyclists were included, only urinary incontinence (UI) symptoms were analysed. Simeone et al. [20], in assessing the prevalence of LUT symptoms in a population of 623 female athletes (305 amateur, 279 competitive level and 39 professional) included 22 female cyclists. In this study, female cyclists were included in the low-impact sports group. Unfortunately, these authors analysed the prevalence of LUT symptoms in relation to the entire study population, and in relation to the female cyclists they only indicated that, as in the case of female football players, urge incontinence symptoms were the most common and that some of the female cyclists had symptoms of dysuria. The latter symptoms were linked to the fact that cycling is classified as an endurance sport, which may lead to dehydration and loss of electrolytes. Unfortunately, this paper did not provide any specific data on the prevalence rate of the aforementioned symptoms among female cyclists with which one could compare the results of our study. These authors suggested that in the case of female cyclists, as in female horseback riders, certain injuries in conjunction with excessive chronic stress in the perineal region might favour incontinence. However, this speculation in relation to female amateur (recreational) horseback riders was not confirmed in our previous study [21]. Several papers discussing the association between women practising sport and incontinence symptoms have also analysed female cyclists. Rodríguez-López et al. [22] found no evidence of UI symptoms among cyclists, but only one cyclist was included in their analysis, and it is not known if the cyclist was a woman, as the analyses of UI symptoms were conducted collectively for both genders. On the other hand, Whitney et al. [23] report that of 31 female cyclists analysed, 19.4% developed UI symptoms during sporting activity. However, these authors do not mention anything about the severity of these symptoms. In a systematic review of 2018 [24], the prevalence of UI symptoms among 89 female cyclists included in the analysis was estimated at 10.11%. The problem of the prevalence of stress urinary incontinence among competitive female athletes was approached slightly differently by Selecka et al. [25], who categorised the competitive sports practised according to the community determinants of their practice and the volitional factors accompanying the sports in question. Cycling with athletics running on middle-distance tracks in this study was included in the functional-mobilisation sports group (group size n = 49). These authors showed that the relative risk of developing stress urinary incontinence was most significant in this group (95% CI: 1.04–3.68; *p* < 0.03), and the averaged stress urinary-incontinence risk factor (OR = 1.96) was significantly higher compared to the other five sport categories. For the strength sports, aesthetic-coordination sports and heuristic collective sports with a hockey stick, the coefficient of relative stress urinary-incontinence development was more than twice as low (OR consecutively = 0.77; 0.69 and 0.63) [25].

In our study, the occurrence of urge urinary incontinence and stress urinary incontinence was assessed separately due to the design of the LUTS questionnaire used for the study. Symptoms of urge urinary incontinence were found in 21 cyclists (27.6%), but in as many as 20 cyclists these symptoms were infrequent, and only 1 cyclist had symptoms sometimes. In contrast, as many as half of the female cyclists surveyed reported stress urinary incontinence symptoms, but in 16 women these symptoms occurred infrequently, in 20 sometimes, and only 2 cyclists reported frequent stress urinary incontinence symptoms. Although the risk of developing symptoms of both urge urinary incontinence and stress urinary incontinence is statistically significantly higher compared to women who do not participate in sport, symptoms occurring in a negligible proportion of the female competitive cyclists surveyed are of clinical significance (2.63%). In clinical terms, pain in the bladder seems to be a greater problem in the studied group of female athletes, the frequent occurrence of which was indicated by twice as many female athletes (5.26%). In the latter case, such a condition may be the result of more frequent lower urinary tract infections associated with direct contact between the perineum and the cycling saddle and exercise-related excessive sweating and overheating of this body area. Gaither et al. [12], in their study, found a statistically significant higher frequency of urinary tract infections in low- and high-intensity cyclists compared to non-cyclist women. This fact was explained by these authors by shorter urethral length in women, and increased surface area contact in conjunction with lesser breathability within the perineum [12].

On the basis of our study, it can be concluded that sitting for hours on the saddle of a professional bicycle associated with strenuous cycling training and high physical exertion during a cycling race has some adverse effects on the condition of the lower urinary tract. This is particularly true for increased frequency of daily micturition, urge urinary incontinence and stress urinary incontinence symptoms and pain in the bladder. The incidence of LUTSs increases with years of cycling and increased hours of cycling training. The existence of a relationship between years of training and hours of training per day in relation to the incidence of UI symptoms in female athletes has also been pointed out by other authors [20,26,27]. The severity of LUTSs among competitive female cyclists is not high and, in the vast majority of cases, is not a clinical problem. It is also possible that competitive cycling will not result in lower urinary tract dysfunction after the end of the sporting career. Indeed, studies have found that many years after the end of a sporting career, there were no differences in UI prevalence both between high-impact and low-impact athletes, and between elite athletes and women with no history of competitive sport [28,29]. An interesting observation from the point of view of cycling coaching is the discovery of a significant association between the occurrence of LUTS symptoms in competitive female cyclists and past natural births. This fact, on the one hand, indicates the necessity for women to carefully plan the timing of their return to the continuation of their sporting career after childbirth, taking into account the assessment of their urological health. On the other hand, it indicates the legitimacy of enriching sports training in these women with individually tailored exercises in the so-called pelvic floor training [30,31,32]. It is also worth considering the implementation of appropriate preventive measures during the pregnancy of competitive female cyclists. Indeed, studies show that women who exercised their pelvic floor muscles during pregnancy were 62% less likely to experience UI in late pregnancy and had a 29% lower risk of UI 3–6 months postpartum [33].

The CLSS questionnaire used in this study has a simple design, is clear, contains specific questions, and appears to be a very good research tool for assessing the prevalence of LUTSs. However, it should be borne in mind that the prevalence of the assessed symptoms is highly subjective, despite the total anonymity of the study. It appears that the results of this type of study may be influenced by the emotional approach of the respondent to the health problems raised in the questions. In some women, the presence of a health problem may be suppressed, based on the attitude “after all, I am young, athletic and healthy and such a problem does not concern me”, or trivialised with the opinion “every woman sometimes has such symptoms and it is not a health problem”. Other women, on the other hand, may see in every trivial and sporadic symptom a serious health problem. Indirectly, this approach to lower urinary tract symptoms may be supported by the results of two studies that combined a subjective (anonymous questionnaire) and an objective (sanitary pad test) method to assess the occurrence of UI incidents. In a study by Da Silva Pereira et al. [34], it was found that female competitive volleyball players rated UI incidence symptoms more objectively than female amateur volleyball players. In contrast, in a study by Dos Santos et al. [27], in a questionnaire survey, 52% of women signalled the presence of UI symptoms during training, while in fact, in a sanitary pad test, in the same group of subjects, leakage was registered in 43.7%. Interestingly, 24% of female athletes who did not describe incontinence symptoms in the questionnaire had urine leakage in the sanitary pad test. Therefore, we cannot have a guarantee that, in the case of competitive female cyclists, all answers were given according to fact, and that some of the results are not ‘false positives’ or ‘false negatives’. Of course, in future studies assessing, for example, the prevalence of UI symptoms in competitive female cyclists, the questionnaire survey could be supported by a sanitary pad test, but it is difficult to imagine a fully objective assessment of the prevalence of all LUT symptoms. 

Another limitation of this work is that the research covered women practising two cycling disciplines—road cycling and cross-country cycling. In road cycling, the duration of training and cycling races may have a potential impact on the lower urinary tract. Their specific nature may include, for example, forced holding of urine during training or cycling competitions. In turn, in cross-country cycling, which involves riding on uneven terrain, very frequent mechanical shocks may have a significant impact on the lower urinary tract. A problem that is not discussed in these studies is replenishing fluids during long-term cycling. The amount and type of fluid intake, including the consumption of alcoholic beverages, was not analysed in a similar manner, which may affect the frequency of micturition during the day and the tendency to nocturia. 

In future research on women practising cycling competitively, the analysis should be carried out separately and divided into different types of cycling. Such analyses should also take into account a wider range of factors that could potentially influence the condition of the lower urinary tract in women practising cycling competitively. Of course, the specifics of long-term cycling in various weather and environmental conditions can potentially have health implications for the entire pelvic floor in women. Therefore, it is worth extending the research to also analyse the occurrence of symptoms from all systems located in the female pelvis. Only then could the potential health risks associated with a long-term stay on the bicycle seat in competitive sport conditions be clearly concluded.

## 5. Conclusions

The results of our study indicate that competitive female cycling is associated with some risk of lower urinary tract symptoms. The most common symptoms among female cyclists are increased daily micturition (55.26%), pain in the bladder (52.63%) and stress urinary incontinence (50%). However, the severity of these symptoms is not clinically significant in the vast majority of the cases. 

Nevertheless, the more frequent occurrence of LUT symptoms in women competitive cyclists indicates the need to include them in periodic medical examinations. 

An important factor influencing the appearance of LUTSs in women who cycle is the history of natural childbirths. This fact should result in modifying the training programs of women returning to competitive sports after giving birth to also include pelvic floor muscle exercises. 

## Figures and Tables

**Figure 1 healthcare-12-01163-f001:**
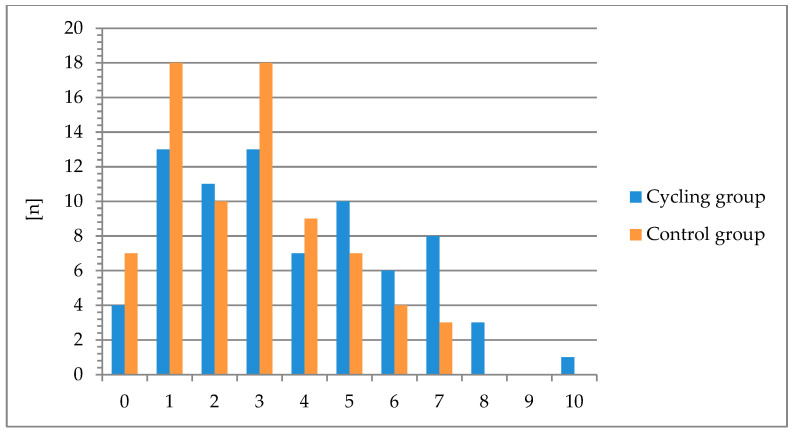
Number of lower urinary tract symptoms among women cyclists and women in the control group.

**Figure 2 healthcare-12-01163-f002:**
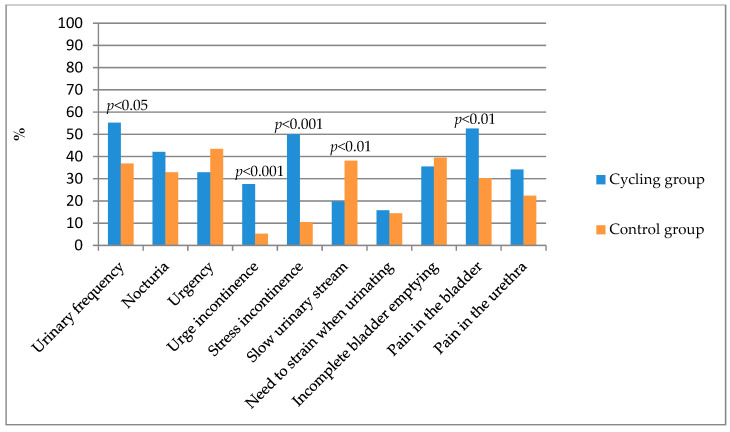
Occurrence of lower urinary tract symptoms according to the groups.

**Figure 3 healthcare-12-01163-f003:**
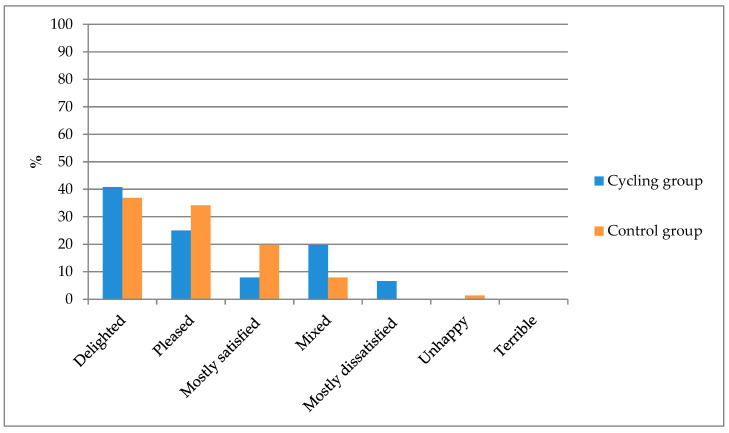
Respondents’ feelings related to the assessment of their current condition of the lower urinary tract (Chi^2^ = 14.956; *p* = 0.021).

**Table 1 healthcare-12-01163-t001:** Demographic data of the participants (mean; SD; range).

Characteristics	Cycling Group (*n* = 76)	Control Group (*n* = 76)	*p* Value
Age (years)	25.07 (4.4) 19–43	25.42 (5.0) 18–40	0.645 ^a^
Weight (kg)	63.5 (5.9) 49–74	63.2 (10.8) 46–103	0.213 ^b^
Height (cm)	170.1 (6.9) 152–188	168.5 (5.4) 155–179	0.130 ^b^
BMI (kg/m^2^)	21.97 (2.2) 18.39–28.08	22.22 (3.5) 15.92–34.42	0.680 ^b^
Natural childbirth (*n*)	17	15	0.618 ^c^
Caesarean section (*n*)	13	7	0.212 ^c^
Sports career (years)	6.29 (4.4) 1–21	-	-
Training frequency (no./week)	4.34 (1.4) 2–10	-	-
Age (years)	12.72 (6.6) 3–30	-	-

^a^ *t*-test; ^b^ U Mann–Whitney test; ^c^ Chi^2^ test.

**Table 2 healthcare-12-01163-t002:** Number of symptoms and level of acceptance of lower urinary tract condition (mean; SD, 95% CI) in Cycling and Control group.

	Cycling Group	Control Group	*p* Value
No. of Symptoms	3.66 (2.3) 3.12–4.2	2.74 (1.9) 2.31–3.16	0.05 ^b^
The level of acceptance of the condition of CLSS	1.26 (1.3) 0.95–1.57	1.04 (1.1) 0.80–1.28	0.583 ^b^

^b^ U Mann–Whitney test.

**Table 3 healthcare-12-01163-t003:** Lower-urinary-tract symptom intensity level and overall CLSS score (mean; SD, 95% CI) in the Cycling and Control group.

Symptom	Cycling Group	Control Group	*p* Value
Daytime frequency	0.75 (0.8) 0.58–0.93	0.46 (0.7) 0.31–0.61	0.01 ^a^
Nocturia	0.45 (0.5) 0.32–0.57	0.41 (0.6) 0.26–0.55	0.683 ^a^
Urgency	0.41 (0.6) 0.26–0.55	0.55 (0.7) 0.39–0.71	0.184 ^a^
Urgency incontinence	0.29 (0.4) 0.18–0.4	0.05 (0.2) 0.001–0.1	0.05 ^b^
Stress incontinence	0.86 (0.9) 0.61–1.03	0.13 (0.4) 0.04–0.23	0.001 ^b^
Slow stream	0.28 (0.6) 0.14–0.41	0.50 (0.7) 0.34–0.66	0.05 ^a^
Straining	0.18 (0.4) 0.08–0.29	0.17 (0.4) 0.07–0.27	0.857 ^a^
Incomplete emptying	0.42 (0.6) 0.28–0.56	0.47 (0.7) 0.32–0.63	0.613 ^a^
Bladder pain	0.79 (0.9) 0.58–0.99	0.36 (0.6) 0.22–0.49	0.01 ^b^
Urethral pain	0.42 (0.6) 0.28–0.57	0.28 (0.6) 0.15–0.40	0.138 ^a^
CLSS Score	4.8 (3.5) 3.99–5.61	3.38 (2.6) 2.79–3.97	0.05 ^b^

^a^ *t*-test; ^b^ U Mann–Whitney test.

**Table 4 healthcare-12-01163-t004:** Correlation between the years of competitive cycling, number of training sessions and training hours per week and severity of lower urinary tract symptoms in female competitive cyclists.

Symptom/CLSS Score	Years of Training	Number of Trainings Per Week	Hours of Trainings Per Week
Urinary frequency	0.264 ^1,^*	0.141 ^1^	0.207 ^1^
Nocturia	0.272 ^1,^*	0.194 ^1^	0.296 ^1,^**
Urgency	0.241 ^1^	0.054 ^1^	−0.156 ^1^
Urge incontinence	0.129 ^1^	0.085 ^1^	0.134 ^1^
Stress incontinence	0.379 ^1,^***	0.247 ^1,^*	0.437 ^1,^***
Slow urinary stream	0.223 ^1^	0.143 ^1^	0.062 ^1^
Need to strain when urinating	0.093 ^1^	0.027 ^1^	−0.068 ^1^
Incomplete bladder emptying	0.177 ^1^	0.059 ^1^	−0.003 ^1^
Pain in the bladder	0.027 ^1^	0.042 ^1^	0.114 ^1^
Pain in the urethra	0.103 ^1^	0.073 ^1^	0.130 ^1^
No. of Symptoms	0.276 ^1,^*	0.195 ^1^	0.276 ^1,^*
CLSS Score	0.364 ^2,^***	0.203 ^2^	0.288 ^2,^**

^1^ Spearman test; ^2^ Pearson test; * *p* < 0.05; ** *p* < 0.01; *** *p* < 0.001.

**Table 5 healthcare-12-01163-t005:** Correlation between type of birth given and the CLSS questionnaire scores.

Symptom/CLSS Score	Cycling Group		Control Group	
	Natural Childbirth	Caesarean Section	Natural Childbirth	Caesarean Section
Urinary frequency	0.339 ^1,^**	0.098 ^1^	−0.109 ^1^	−0.198 ^1^
Nocturia	0.427 ^1,^***	−0.111 ^1^	0.029 ^1^	0.047 ^1^
Urgency	0.146 ^1^	0.100 ^1^	0.022 ^1^	0.227 ^1^
Urge incontinence	0.304 ^1,^**	−0.132 ^1^	0.135 ^1^	0.034 ^1^
Stress incontinence	0.517 ^1,^***	−0.137 ^1^	0.228 ^1,^*	0.018 ^1^
Slow urinary stream	0.436 ^1,^***	−0.038 ^1^	−0.133 ^1^	0.191 ^1^
Need to strain when urinating	0.326 ^1,^**	0.026 ^1^	−0.119 ^1^	−0.085 ^1^
Incomplete bladder emptying	0.348 ^1,^**	−0.016 ^1^	−0.150 ^1^	−0.044 ^1^
Pain in the bladder	0.145 ^1^	0.145 ^1^	−0.137 ^1^	0.031 ^1^
Pain in the urethra	0.081 ^1^	−0.196 ^1^	−0.150 ^1^	0.026 ^1^
No. of Symptoms	0.542 ^1,^***	−0.010 ^1^	−0.131 ^1^	0.049 ^1^
CLSS Score	0.569 ^2,^***	−0.067 ^2^	−0.126 ^2^	0.068 ^2^

^1^ Spearman test; ^2^ Pearson test; * *p* < 0.05; ** *p* < 0.01; *** *p* < 0.001.

## Data Availability

The data that support the findings of this study are available from the corresponding author upon reasonable request.

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
