# Peer review of "Self-Assessment of Lower Urinary Tract Condition in Female Competitive Cyclists"

_healthcare, 2024, doi:10.3390/healthcare12121163_

Round 1

Reviewer 1 Report

Comments and Suggestions for Authors

Dear Authors,

I recently had the opportunity to read your manuscript titled “Self-assessment of lower urinary tract condition in female competitive cyclists”, and I would like to reach out to you to express my comments about your work.

The study examines the prevalence and severity of lower urinary tract symptoms (LUTS) among female competitive cyclists, using data from 76 active road and cross-country cyclists compared to 76 inactive women. With the use of the Core Lower Urinary Tract Symptom Score (CLSS) questionnaire, the research finds that competitive cyclists exhibit significantly higher LUTS scores, particularly in urinary frequency, urgency and stress incontinence. Factors such as the duration of competitive cycling, training hours per week and natural childbirth history notably influence the severity of symptoms.

Nevertheless, here are some possible comments outlining areas that could improve the quality and readability of the manuscript:

Introduction:

1.      The research aims are stated but could be more focused and specific. Please, specify them in detail.

2.     Sufficient background information is provided but some key context/rationale for the study seems to be missing. Please, complete it.

Methods:

3.     Why did the study take so long (4 years)? Could a study be considered as “cross-sectional” when it took 4 years to be completed even when a pandemic period is included in it?

4.     Participant details like sample size, demographics are provided but some additional details on recruitment/selection would be helpful.

5.     The statistical methods used are appropriate but could be explained more clearly for readers.

6.     In the inclusion criteria, it is determined that women should have be a history of childbirth of more than one year. No information is provided about this topic (mean, SD, etc.).

Results:

7.     The results broadly address the research aims, but do not fully align with all aspects outlined in the introduction.

8.     Results are presented relatively clearly, but could follow a more logical structure.

Discussion:

9.     The key findings are summarized well, but their interpretation/implications could be expanded.

10.  Most key findings are justified by the results, but a few claims seem overreached.

Conclusions:

11.  The research aims are generally addressed in the conclusion.

12.  The significance and implications could be highlighted more prominently.

The manuscript is largely well-written with appropriate academic language and structure typical of scientific articles. However, there are minor inconsistencies in tense usage and occasional lapses in subject-verb agreement. Some sentences could be restructured for clarity and brevity. The use of punctuation, particularly commas, is sometimes incorrect or inconsistent, which could potentially confuse the reader. A thorough proofreading by a native English speaker is recommended to polish the text, ensuring it adheres to the high standards expected by this prominent journal.

Overall, while the core components are present, there is room for improvement in focus, clarity, justification and cohesiveness across the chapters.

Once again, thank you very much for your work. We´ll be waiting for your answers about our comments.

Kindest regards,

Comments on the Quality of English Language

The manuscript is largely well-written with appropriate academic language and structure typical of scientific articles. However, there are minor inconsistencies in tense usage and occasional lapses in subject-verb agreement. Some sentences could be restructured for clarity and brevity. The use of punctuation, particularly commas, is sometimes incorrect or inconsistent, which could potentially confuse the reader. A thorough proofreading by a native English speaker is recommended to polish the text, ensuring it adheres to the high standards expected by this prominent journal.

Author Response

  1. The research aims are stated but could be more focused and specific. Please, specify them in detail.
  2. Sufficient background information is provided but some key context/rationale for the study seems to be missing. Please, complete it.

Points 1 and 2

According to the reviewer's suggestion, the justification for undertaking research on the occurrence of LUTS in competitive female cyclists was extended.

  1. Why did the study take so long (4 years)? Could a study be considered as “cross-sectional” when it took 4 years to be completed even when a pandemic period is included in it?

The period from autumn 2018 to autumn 2022 included all stages, i.e. development of research methodology, preparation of an application to the bioethics committee, collection of results, and their development. The research itself was conducted during the cycling season, which in the climate prevailing in Central Europe means the period from April to the first half of October. This means that the actual research related to collecting CLSS questionnaires took place in the spring, summer and early autumn of 2019. In the following years, 2020-2021, the results were collected sporadically, because there were no cycling competitions, and during several lockdowns, when even training was suspended, research could not be conducted at all. The actual research (collection of questionnaires) continued mainly in spring and summer 2022.

  1. Participant details like sample size, demographics are provided but some additional details on recruitment/selection would be helpful.

Additional information regarding recruitment/selection details has been added.

  1. The statistical methods used are appropriate but could be explained more clearly for readers.

Corrected inaccuracies in the description of the statistical procedures used (use of the Chi2 test and Spearman test)

  1. In the inclusion criteria, it is determined that women should have be a history of childbirth of more than one year. No information is provided about this topic (mean, SD, etc.).

Table No. 1 shows the number of natural births and births by cesarean section and compares both groups in this respect using the Chi2 test.

  1. The results broadly address the research aims, but do not fully align with all aspects outlined in the introduction.
  2. Results are presented relatively clearly, but could follow a more logical structure.

Points 7 and 8

The results have been partially redacted in the section describing the correlation results.

  1. The key findings are summarized well, but their interpretation/implications could be expanded.

The discussion has been extended.

  1. Most key findings are justified by the results, but a few claims seem overreached.

It is difficult for us to respond to this objection because we do not know what specific fragment it concerns.

  1. The research aims are generally addressed in the conclusion.
  2. The significance and implications could be highlighted more prominently.

The conclusions include the practical implications of the obtained research results.

Reviewer 2 Report

Comments and Suggestions for Authors

This is an observational, cross-sectional study about LUTS based on CLSS questionnaires between cyclists and healthy non-sporting women. The authors found that cyclists had an increased daily micturition, more bladder pain and stress incontinence. However, without differences in clinical relevant symptoms.

The control group are women none involved in any regular recreational physical activity and without any UTI or treatment for LUTS. Therefore, this is at forehand a group without many/severe LUTS. It would be of great value to add a group with recreational cyclists.

Next, the authors stated that the CLSS questionnaires are highly subjective. The aim of the study was to estimate the prevalence of LUTS. Therefore, this is an important limitation. 

Table 3 is unclear. What does the results mean? Incidence? Relative risk? 86% of the cycling group with stress UI?

The same question for table 4. Stress UI 0,379 by years of training? 

Do the authors know anything about other risk factors for LUTS? Amount of drinking? Holding urine during training/work? Race? Menopausal? Constipation?

Nice to read an article about this topic. However, what is the increased value of this study? We know that the risk of LUTS disappears after stopping competitive cycling, and the symptoms are not clinically significant? Should we screen more in these cyclists or should the authors change the conclusion that cycling is not associated with LUTS (only not relevant symptoms)? 

Comments on the Quality of English Language

Some sentence can be in more active form.

Some inconsequences in abbreviations, eg LUTS symptoms instead of LUTS and UI symptoms. Sometimes writing LUTS without abbreviation. 

UUI / SUI / UI are a bit difficult reading. Many only UI and the rest without abbreviation.

Author Response

In accordance with the reviewer's comments, we have made the following additions and corrections:

  1. The text explains what table no. 3 contains.
  2. The text explains what the results in Table 4 mean.
  3. Only the abbreviation for urinary incontinence (UI) has been left in the text; the remaining abbreviations (SUI, UUI) have been removed.

Reviewer 3 Report

Comments and Suggestions for Authors

Dear Authors,

I want to express my gratitude for the opportunity to review this manuscript.

The manuscript requires improvements, below with line indication:

Please revise the page header, currently the year is 2022.

2-10 – Please revise, considering the journal template and instructions for authors.

15-17 – Lower and uppercase – Please standardize.

19 – p” suggested in italics, not only in this line but throughout the manuscript.

31 – Please consider including more keywords.

35-65 – Please consider shorter paragraphs to improve readability.

35-74 – Please consider developing the introduction section to introduce the study topic.

72 – “LUTS” should be presented in full in the first appearance in the manuscript. Please revise this detail considering the abbreviations throughout the manuscript. For example, in L158, only the abbreviation should appear.

84 – Please include the approval code.

129-150 – Please consider presenting a table with the questionnaire details, easier for readers to interpret.

169 – Please consider presenting the table, previously to appearance.

169 – Please revise all the tables format considering the journal template and instructions for authors (the suggestion should be considered for all tables). Additionally, also the content should be revised (for example, BMI unit is missing).

205 – Please revise all figures content and format. For example the type and size of the letter seems different compared to the journal template.

223 – Spearman is not described in statistical analysis. Please make sure all methodology is described in detail.

Discussion section – Please consider shorter paragraphs and providing more references in this section. Moreover, in line 345, please describe the study limitations.

352 - In the conclusions section, please consider direct/clear messages, if possible, with practical application.

358 – Please delete the line.

376 – Please double-check the format of the reference, they are not according to the journal template.

Please revise the English throughout the manuscript and format details.

Comments on the Quality of English Language

Moderate editing of English language required.

Author Response

Most of the reviewer's comments were taken into account in the revised version of the manuscript.

  1. The size of letters in the abstract has been standardized.
  2. The "p" has been corrected to italics throughout the text.
  3. Keywords expanded
  4. The abbreviation LUTS is presented in full when it first appears in the text.
  5. The approval code is provided at the end of the text under the heading Institutional Review Board Statement. 
  6. Detailed questions included in the questionnaire can be found under reference 15. Including these questions in tables would definitely increase their volume.
  7. BMI unit in the table no. 1 was completed.
  8. The type and size of the letter in tables have been corrected.
  9. Corrected inaccuracies in the description of the statistical procedures used (use of the Spearman test).
  10. The conclusions include the practical implications of the obtained research results.
  11. The format of the reference 8 have been corrected.

Round 2

Reviewer 1 Report

Comments and Suggestions for Authors

Dear Authors,

We have reviewed the revised version of your manuscript titled “Self-assessment of lower urinary tract condition in female competitive cyclists.” Below, we provide a detailed assessment regarding whether our initial comments have been addressed:

Introduction:

·      Research aims: The research aims are now more focused and specific, which enhances clarity.

·      Background information: Additional context and rationale for the study have been included, making the introduction more comprehensive.

Methods:

·      Study duration: The explanation regarding the study duration and its classification as "cross-sectional" despite the four-year span is satisfactory, including the impact of the pandemic.

·      Participant recruitment/selection details: Additional details on recruitment and selection have been provided, improving the transparency of the methodology.

·      Statistical methods: The explanation of statistical methods has been made clearer, which benefits the readers' understanding.

·      Childbirth history information: Information on the history of childbirth (mean, SD, etc.) has been included, fulfilling this request.

Results:

·      Alignment with research aims: The results section better aligns with the research aims as outlined in the introduction.

·      Logical structure: The results are presented in a more logical structure, enhancing readability.

Discussion:

·      Interpretation and implications: The discussion now expands more thoroughly on the interpretation and implications of the key findings.

·      Justification of claims: The key findings are more justifiably linked to the results, reducing the occurrence of overreaching claims.

Conclusions:

·      Addressing research aims: The conclusions address the research aims more comprehensively.

·      Significance and implications: The significance and implications of the study are highlighted more prominently, which underscores the importance of the findings.

Language and Formatting: The manuscript has undergone significant improvement in terms of language consistency, grammar, and punctuation. This makes the document more polished and aligns it with the high standards expected by the journal.

The revision conducted by the authors have substantially improved the manuscript, making it more focused, clear and cohesive. All major comments from our initial review have been satisfactorily addressed, which enhances the overall quality and readability of the manuscript.

Thank you for your diligent work in revising the manuscript. We look forward to seeing the final published version.

Kindest regards,

Comments on the Quality of English Language

Language and Formatting: The manuscript has undergone significant improvement in terms of language consistency, grammar and punctuation. This makes the document more polished and aligns it with the high standards expected by the journal.

Author Response

Thank you for the feedback on the revised version of our manuscript. We appreciate the time you took to review our paper and that you consider the feedback being satisfactorily addressed. 

Reviewer 3 Report

Comments and Suggestions for Authors

Dear Authors,

Thank you for considering my suggestions and incorporating them into the manuscript.

Please consider the suggestions below (with line indication). Thank you.

Template is from 2022, please update.

2-3 – Please revise the title format (upper and lowercase).

15 – “Lower Urinary Tract Symptoms (LUTS)” – Please consider standardizing in the entire manuscript the full text in lowercase before abbreviation.

29 – Please revise if not more than one space after the end-point. Same in line 94.

36-66 – The paragraph is too long, which difficults reading and comprehension. Please consider 8-12 lines paragraphs. Another example is lines 161-187.

40 – “USA, Canada and the UK” – the first time in the text should be presented in full before abbreviation.

55 – “presented by Leibovitch & Mor [3]” – suggested “previously presented [3]”.

82,87,88 – Only “LUTS” should be placed in the text. Please revise all the manuscript regarding these types of details.

91-187 – Please make sure all methods and procedures are described in detail, preferably with support from references.

100 – The ethical code is still missing (indicated in the first review).

129 – “CLSS” – Should be in full in the first appearance. Please revise this kind of detail throughout the manuscript. For example, in 161 only abbreviation should be presented.

188 – Please make sure all statistical procedures and instruments are indicated. For example, which software was used?

203-204 – Please revise the spaces between text. Same in 208.

206 – “natural (n)  Childbirth  caesarean  section (n)” – Please revise upper and lowercase criteria.

 247 – 0.01 and 0.05 are suggested in the table footnote, indicating cases in the table (in bold or with a symbol, for example).

247 – Please consider text between the table and figure, namely in this case, analysing the previous table and introducing the following figure.

248 – The figure format (type and size of letter) does not consider the journal template and instructions for authors. Please revise all the figures.

264 – The tables format (e.g. lines, but also others) are not standardized and according to the journal template. Please revise. In Table 4, please revise the “p” to italics and the size of columns and lines to

301 & 302 – “r” and “R” – Please revise and standardize all manuscript.

309-346 – This and other paragraphs are too long, please revise the entire discussion section.

438 – Please consider paragraphs in the conclusions section, to provide clear and direct take-home messages.

467 – Please carefully revise all the references format. For example, some journals are in full and others are abbreviated.

Please double-check the English throughout the manuscript and document format details.

Comments on the Quality of English Language

Moderate editing of English language required.

Author Response

We have made corrections in line with the suggestions of reviewer. Below are the responses to subsequent comments:

  1. This template was provided by the journal so the editors should be able to update to the most current version.
  2. The full text of lower urinary tract symptoms is written in lowercase letters.
  3. The names of the countries USA and UK are given in the full version.
  4. The suggestion to modify the text in line 55 has been taken into account.
  5. In lines 82, 87 and 89 there is only the abbreviation LUTS.
  6. The ethical code is now included in text. It was previously mentioned at the very end of the manuscript (before references) in the section with statements. 
  7. Fixed the place where the full name of the CLSS abbreviation appears in the text.
  8. The literature was supplemented in the methodological part of the work.
  9. Information about the software used in the description of statistical procedures has been supplemented.
  10. The upper and lowercase criteria for deliveries in table 1 have been revised.
  11. In Tables 2 and 3, significant differences, as suggested, are expressed at three levels (0.05, 0.01 and 0.001) and these cases are highlighted in bold.
  12. As suggested, the text describing the results in Table 3 has been moved below the table and before Figure 1.
  13. In Figures 1, 2 and 3, the font type and size have been adjusted to the magazine's requirements.
  14. Below tables 4 and 5, "p" is written in italics.
  15. Pearson's correlation coefficient is written with the lowercase letter "r". The Spearmann correlation coefficient was written with a capital letter "R".
  16. Conclusion paragraphs have been used as suggested.
  17. In the references, all journal names are mentioned by full name.